# Physical and Chemical Properties of Limestone Quarry Technosols Used in the Restoration of Mediterranean Habitats

Pau Solé [1,2], Diana Ferrer [1], Irene Raya [1], Meri Pous [3], Robert Gonzàlez [3], Sara Marañón-Jiménez [1,4], Josep Maria Alcañiz [1,4] and Vicenç Carabassa [1,4,*]

1 CREAF, E08193 Bellaterra (Cerdanyola del Vallès), Catalonia, Spain; pau.sole@irta.cat (P.S.); i.raya@creaf.uab.cat (I.R.); josemaria.alcaniz@uab.cat (J.M.A.)
2 IRTA—Mas Badia (La Tallada d'Empordà), Catalonia, Spain
3 Agència de Residus de Catalunya, Generalitat de Catalunya. Catalonia, Spain
4 Universitat Autònoma de Barcelona, E08193 Bellaterra (Cerdanyola del Vallès), Catalonia, Spain
* Correspondence: v.carabassa@creaf.uab.cat; Tel.: +34-935-813-355

**Abstract:** The lack of topsoil is frequently a limiting factor in limestone quarry restoration. This implies that new technosols for maintaining target habitats must be created using mining wastes as the main components. We designed three different technosols using different combinations of mineral materials (mining wastes, excavated soils and topsoil), organic amendment types (compost and digestate) and doses for the restoration of target habitats. Moreover, we monitored the main physicochemical indicators of the quality of the technosols. We observed not only an increase in soil organic carbon and plant nutrients related to the application of any type of organic amendment, but that the digestate mostly increased the soil resistance to erosion by improving soil aggregation even before the emergence of vegetation. Soil-water-retention capacity only improved in technosols built with organic amendments and topsoil. The combination of mining wastes, organic amendments and a superficial horizon of topsoil resulted in the most optimal technosol for the restoration of limestone quarries in the Mediterranean climate.

**Keywords:** soil rehabilitation; habitat restoration; organic amendments; compost; digestate; soil properties

## 1. Introduction

The extractive industry plays an essential role in modern society, as it supplies minerals, aggregates and other materials to fundamental economic sectors, such as construction or industry. However, open-cast mining causes localised and drastic land degradation to the Mediterranean basin, where extensive karst landscapes have been exploited for centuries [1]. Therefore, these extractive activities must rely on restoration plans according to Catalan and European regulations [2,3], which usually require the exploitation entity to deposit a guarantee bond that can only be recovered upon completion of the restoration.

Land restoration affected by extractive activity must be conducted according to a previously approved project, which essentially begins with the geomorphological remodelling of the relief. For this purpose, slope–berm models have been traditionally used [4]. This step must consider the creation of an associated drainage network that enables runoff conduction. Subsequently, the edaphic cover must be replaced, and in the last stage, sowings and plantations with autochthonous species may be carried out to control erosion and restore the native ecosystem in case natural colonisation is not developing fast enough [5].

In an ideal situation, the edaphic cover is replaced by using former soils in the area that were previously separated by layering and stored in one-meter-high piles to avoid compaction. But, if conserving the soil prior to exploitation is not possible, or

the soil is not found in sufficient quantities, we have to seek viable alternatives. This problem can often be solved by preparing artificial soils using mine wastes combined with appropriate amendments. These soils are classified as technosols, according to the World Reference Base (WRB) for Soil Resources [6]. Extractive activities can use excavated soils and/or residues from the quarry extraction processes or crushed rocks to construct these technosols [7]. Excavated soils are natural soils obtained off-site that are transported to the quarry, mostly from nearby works, where the quarry materials are also destined to go. The composition of such soils is highly variable, and may include so-called artefacts (rubble, iron, ceramics, plastics, etc.). Excavated soils can also come from large public works, such as highway or railway constructions; thus, the soil composition can be relatively homogeneous. Mining wastes, which are residues from extraction and aggregate processing (blasting debris, production waste) in the limestone quarry or mining area, are usually available in large quantities for constructing technosols. These wastes are normally used to restore the topography, and can also be used as coarse fraction to emulate the natural stoniness or as a base of mineral material in technosols. In the latter case, the properties of such wastes need to be improved by adding excavated soils and/or organic amendments from separate-at-source municipal organic wastes [8]. With this approach, the construction of technosols can contribute to the valorisation of organic and mineral wastes in the context of circular economy. Moreover, the valorisation of mining wastes also contributes to reducing transport costs, and thus, greenhouse gas emissions.

Although not strictly necessary, adding organic amendments to technosols is highly recommended, as these amendments provide essential nutrients for plants, enhance biological activity and can increase the soil-water-retention capacity (WRC) [9]. Organic materials can also significantly accelerate vegetation growth if they have adequate C:N ratios as well as phosphorous and potassium contents. Rapid vegetation growth is especially important for controlling soil erosion [10]. However, as organic amendments with relatively low stability and labile organic substrates, such as sewage sludge, manure or biostabilised substrates, can quickly generate a surplus of mineral nitrogen in the soil that can leach to the groundwater, their effects must therefore be closely monitored [11].

The use of organic amendments in constructing technosols is widely studied as an alternative to restoring heavily degraded soils, reducing deficiencies in physical properties (water retention capacity, compaction, resistance to erosion, among others), organic matter contents and nutrients [12–14].

Soil biology plays a critical role in the utilisation of organic amendments in technosols. The soil microbiota, including bacteria, fungi and other microorganisms, are responsible for breaking down organic matter into nutrients that can be used by plants. Utilising organic amendments also stimulates the growth of soil microbiota [15].

Although there have been extensive studies on the effects of technosols on soil properties and the provision of ecosystem services over the years, there are still insufficient studies focused on the design and characteristics of technosols specifically tailored to the restoration of Mediterranean habitats.

The purpose of this work is to determine the effects of different mixtures of mineral and organic wastes used for constructing technosols on soil properties and evaluate their suitability for habitat restoration after an extractive activity.

## 2. Materials and Methods

### 2.1. Experimental Site

This study was carried out in the La Falconera limestone quarry, Barcelona (Catalonia, NE Iberian Peninsula), which produces stones and aggregates for concrete manufacture (Figure 1). The edaphoclimatic conditions in the zone are xeric and thermic [16].

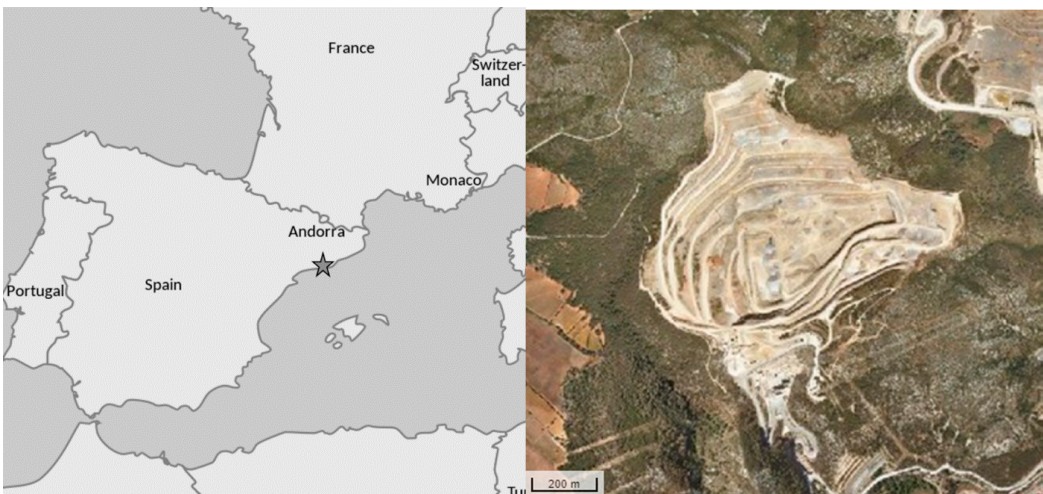

**Figure 1.** (**Left**): Location of the La Falconera quarry in Catalonia, NE Spain. (**Right**): Orthophoto of the La Falconera quarry.

*2.2. Experimental Setting*

The natural soils of the area are very shallow with plenty of rock outcrops, which means there is a scarcity of topsoil for restoration in the quarry. The technosols in this case study were constructed mainly using mineral debris from the quarry, as well as excavated soils at different proportions (Table 1). The characterisation of the materials used can be found in Table 2.

**Table 1.** Proportions in volume (%) of the mineral materials and amendments in $m^3$/ha used for constructing experimental plots.

| Technosol | Blasting Debris (BD) | Production Waste (PW) | Excavated Soils | Amendment Doses ($m^3$/ha) | |
|---|---|---|---|---|---|
| | | | | Digestate | Compost |
| A | 30 | 40 | 30 | 150 | 100 |
| B | 60 | 40 | 0 | 150 | 100 |
| C | 40 | 40 | 20 | 275 | 175 |

Blasting debris is a very heterogeneous and coarse–stony sterile material derived from the limestone extraction process with clay intercalations, <30% fine fraction (<2 mm), a high carbonate content (>60%) and a high presence of large blocks (up to 1.5 m in diameter). Production waste from the aggregate crushing and screening plant was also used. Being essentially gravel, it is a very stony, sterile material that is highly size-classified with little or very little fine fraction (<20%) and has a very high carbonate content (>80%).

Excavated soils were also used for the technosols A and C. These are soils from work cuts. Usually of good quality, they may, however, contain artefacts and have a different mineral composition than the topsoil from the quarry area. Additionally, a small amount (ca. 60 $m^3$, a negligible percentage of the total mineral material) of topsoil recovered from the zone of the quarry before exploitation was used in technosol B. This is a material with a relatively high organic matter content (2.2%), little fine fraction (<20%) and a high carbonate content (60%). Topsoil was stored in a dumping site inside the limits of the quarry, not following the basic prescriptions for soil storage, since piles were higher than one meter.

The natural ecosystems of the quarry prior to the exploitation mostly consisted of Mediterranean Maquis shrublands and some *pinus halepensis* plantations on the lower parts of the terrain.

**Table 2.** Characterisation of the materials used in the construction on technosols in each of the study plots.

| | Units | Blasting Debris (BD) | Production Waste (PW) | Excavated Soils | Topsoils |
|---|---|---|---|---|---|
| 250–75 mm | % | 31 | 0 | 6 | 21 |
| 75–10 mm | % | 34 | 29 | 24 | 34 |
| 10–2 mm | % | 16 | 42 | 23 | 26 |
| <2 mm | % | 19 | 28 | 46 | 19 |
| Bulk density | t/m$^3$ | 1.7 | 1.4 | 1.5 | 1.6 |
| Sand | % * | 45.8 | 67.3 | 63.7 | 38.1 |
| Loam | % * | 20.2 | 15 | 20.6 | 28.4 |
| Clay | % * | 34 | 17.7 | 15.7 | 33.5 |
| Carbonates | % * | 57 | 73 | 18 | 60 |
| pH | * | 8.8 | 9 | 8.7 | 8.8 |
| O.M. | % * | <0.5 | <0.5 | <0.5 | 1.7 |
| E.C. 1:5 | dS/m * | 0.24 | 0.22 | 0.48 | 0.18 |
| N Kjeldahl | % * | 0.08 | 0.073 | 0.12 | 0.1 |
| N-NO$_3^-$ | mg/kg * | 2.7 | 4.1 | 12 | 28 |
| P Olsen | mg/kg * | <5 | <5 | 32.3 | <5 |
| Potassium | mg/kg * | 133 | 35 | 276 | 102 |
| Calcium | mg/kg * | 6319 | 6092 | 5285 | 6481 |
| Magnesium | mg/kg * | 293 | 102 | 269 | 121 |
| Sodium | mg/kg * | 153 | 53 | 167 | 24 |
| Cadmium | mg/kg * | <0.5 | <0.5 | <0.5 | <0.5 |
| Copper | mg/kg * | 28 | <20 | 53 | <20 |
| Nickel | mg/kg * | 42 | 9.8 | 32 | 21 |
| Lead | mg/kg * | 14 | <5 | 29 | 10 |
| Zinc | mg/kg * | 73 | <25 | 100 | 37 |
| Mercury | mg/kg * | <0.4 | <0.4 | <0.4 | <0.4 |
| Chromium | mg/kg * | 66 | 13 | 33 | 30 |

* The characterisation refers to the fine fraction (<2 mm).

After restoring the La Falconera limestone quarry, the target ecosystem is intended to have a mosaic of three different habitats (forest, shrubland and grassland). Therefore, three technosol soil profiles with different compositions and depths were designed in order to meet the specific requirements of the target vegetation (Figure 2). The resulting soils can be classified as Technoleptic Spolic Technosol, according to the WRB [6].

In order to determine the most optimal option for the soil restoration of the quarry, three different treatments of organic amendments were applied to a 10 × 10 m plot with each of these technosols, resulting in nine 10 × 10 m plots. First, soils were laid onto a layer of mining debris that previously filled a 10 m high embankment. The A soil type was the shallowest, with a total thickness of 40 cm, the B soil type had a total depth of 50 cm and the C soil type was the deepest with a total depth of 60 cm. Then three different treatments were specifically created for each type of technosol, two with organic amendments, i.e., compost (COM) or digestate (DIG), and one control (CNT). The organic amendments were mixed with the mineral substrates at different depths according to Figure 2.

The compost and the digestate used were both obtained from the urban waste of the municipality of Granollers (Barcelona). The digestate was produced from source-separated organic household waste, and the compost was produced from this digestate by adding pine wood splinters as a structuring agent. Both materials had low-heavy-metal contents, making them suitable for agricultural uses according to Spanish regulations [17] (Table 3).

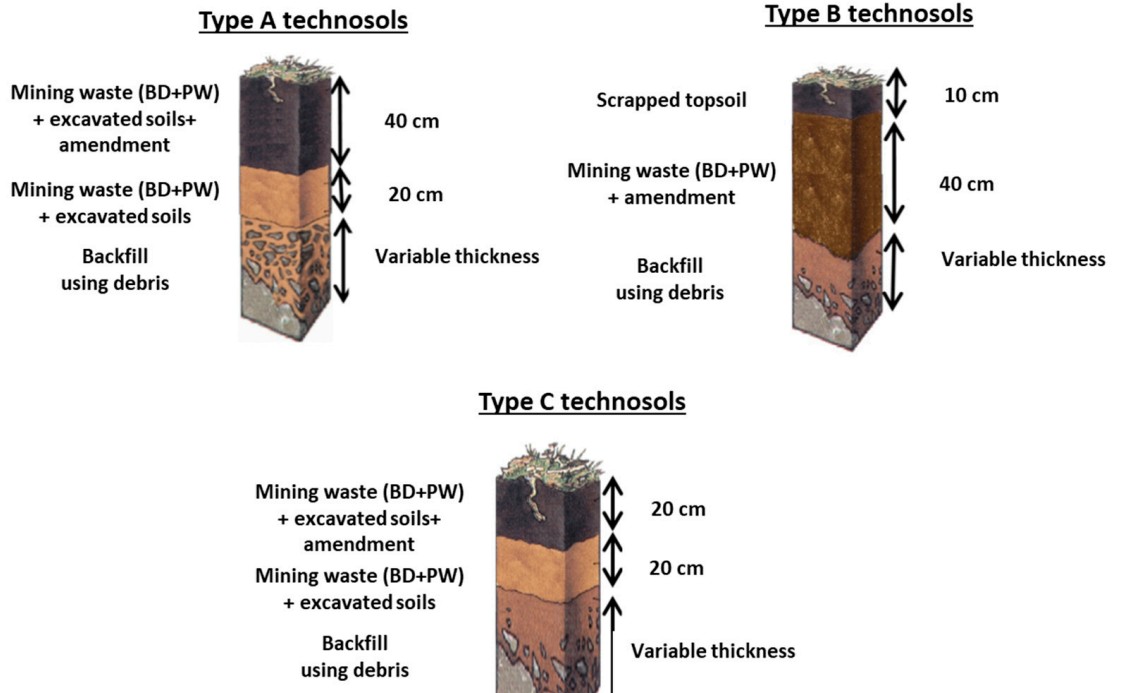

**Figure 2.** Soil profiles of the three types of technosols used in the quarry restoration.

**Table 3.** Analytical characterisation of the organic amendments used in the treatments.

|  | Units | Compost (COM) | Digestate (DIG) |
|---|---|---|---|
| pH |  | 7.9 | 8.7 |
| Electrical conductivity at 25 °C | dS/m | 3.46 | 2.92 |
| C:N ratio |  | 11.6 | 10.9 |
| Organic carbon | % d.m. | 24.6 | 29.3 |
| Stability grade | % | 59.6 | 60.3 |
| $N-NH_4^+$ | % d.m. | 0.12 | 1.27 |
| Total nitrogen | % d.m. | 2.63 | 3.91 |
| BOD | mg $O_2$/g d.m. | 10.26 | 34.20 |
| Phosphorus ($P_2O_5$) | % d.m. | 1.04 | 0.858 |
| Potassium ($K_2O$) | % d.m. | 1.03 | 0.820 |
| Cadmium | mg/kg d.m. | 0.76 | 0.56 |
| Copper | mg/kg d.m. | 105 | 77.3 |
| Chrome | mg/kg d.m. | 25.8 | 20.7 |
| Mercury | mg/kg d.m. | <0.4 | <0.4 |
| Nickel | mg/kg d.m. | 16.7 | 10.7 |
| Lead | mg/kg d.m. | 47.3 | 34.8 |
| Zinc | mg/kg d.m. | 281 | 225 |
| Impurities (Metals + Glass + Plastics) >2 mm | % d.m. | 00.11 | 0.44 |

The technosols were constructed using the machinery available in the quarry (wheel loader and dumpers). For each technosol, mineral wastes were put in piles according to established (Table 1) and mixed proportions. Part of these mineral mixtures were amended by adding predetermined doses of compost or digestate to the piles (Table 1). Amended and unamended mineral mixtures were disposed at different depths, according to Figure 2. Six months after creating the treatments, plots were sown with autochthonous plants from each of the three target habitats.

*2.3. Soil Sampling*

Soil samples of the upper 20 cm layer were taken from 20 × 20 cm squares at two sampling times in order to sample only the amended horizons in all the technosols. The first sampling took place just after spreading the technosols, and the second one right before sowing and six months after the technosol spread. For each sample, particle size was determined by the Robinson pipette method of sedimentation analysis [18]. Equivalent $CaCO_3$ was determined using the Bernard calcimetry method by measuring the $CO_2$ volume released after the HCl addition. Electrical conductivity was determined in a 1:5 (w:v) water extract. Soil organic carbon (SOC) content was obtained by acid dichromate oxidation. Nitrogen content was determined using the Kjeldahl method, available phosphorous using the Olsen method, available potassium using the ammonium acetate extraction method [19], together with other measurements (magnesium, sodium, zinc, cadmium, chrome, lead, mercury, nickel, copper and soil texture) that were carried out in an external EU-accredited laboratory applying standard methods. Before sowing, the infiltration rate of the soil was measured using a double-ring infiltrometer, the penetration resistance was measured using a hand-penetrometer and the soil bulk density was measured by excavating a 20 × 20 × 20 cm hole and filling it with calibrated sand of a known density [20]. Aggregate stability was determined by wet sieving [21] and soil water retention capacity was estimated by gravimetry, using measures of water retention at saturation, after one day, and then after 10 days, these being used as estimates of soil at FC and PWP.

*2.4. Data Analysis*

The effect of the organic amendment treatments on soil properties (aggregate stability, penetration resistance, water retention, E.C., SOC and pH) was tested using one-way ANOVAs for each technosol type, whereby the organic amendment treatment was considered a fixed factor. To test the variables measured at several sampling times, the effect of the organic amendment treatment, the sampling time and the interaction treatment time was tested using repeated-measures ANOVAs (rmANOVAS). The differences among treatments in the soil type or sampling time were further tested by post hoc tests applying the Tuckey correction for multiple testing. Data were transformed when required to improve normality and homoscedasticity. A $p = 0.05$ was used as the cut-off for statistical significance throughout the case study and in this paper. All the results are presented as means ± standard deviation.

**3. Results**

In the general characterisation of the soils performed after the application of the amendments (Table 4), nonstatistically tested changes in nutrients among the treatments can be observed. In A and C technosols, the results show that treatments with compost and digestate increased the nitrogen content in the soil, as expected. Other nutrient (P, K, Mg, Na, Zn) concentrations appear to increase with the application of amendments, while the soil C/N ratio appears to decrease. The metal concentration of the soils does not reach abnormal levels in any of the plots.

The effect of the amendments on the pH and EC is different in each soil (Tables 5 and 6). In soil A, both amendments present a significant pH decrease after the application of the amendments, as well as a significant EC increase. However, after six months, both the pH decrease and the EC increase can only be observed in the digestate treatment. In soil C, both amendments also show a pH decrease and an EC increase after the application, wherein the pH decrease and the EC increase are significantly more remarkable in the digestate than in the compost. After six months, no effect of the amendment treatments was observed in soil C. Soil B exhibited no significant effects of the amendments on its pH or EC.

**Table 4.** Bulk density, granulometric proportions, textural classes and macro- and micronutrients and metal contaminants of the soils in each technosol type and organic amendment treatment. The value for coarse elements refers to the percentage of the original sample. The remaining granulometries are indicated as the percentage of the fraction < 2 mm. Treatments are identified as CNT = control, DIG = digestate and COM = compost. The soil textural classes are abbreviated as ClLo = ClayLoam, SaClLo = SandyClayLoam, Lo = Loam and SaLo = SandyLoam. For each parameter in this table, *n* = 1.

| Soil Parameter | Units | A | | | B | | | C | | |
|---|---|---|---|---|---|---|---|---|---|---|
| | | COM | DIG | CNT | COM | DIG | CNT | COM | DIG | CNT |
| Bulk density | (Mg/m$^3$) | 1.66 | 1.91 | 1.59 | 1.74 | 1.75 | 1.28 | 1.4 | 1.25 | 1.3 |
| Coarse elements (>2 mm) | (%) | 69.3 | 63.7 | 67.2 | 64.2 | 72.4 | 71.2 | 64.8 | 65.6 | 68 |
| Clay | (%) | 25.8 | 23.7 | 29.1 | 35.4 | 33.5 | 33.9 | 22.9 | 18.9 | 29.3 |
| Loam | (%) | 30.7 | 26.8 | 26.3 | 28.3 | 22.4 | 24.8 | 28.5 | 26.5 | 22.2 |
| Sand | (%) | 43.5 | 49.5 | 44.6 | 36.3 | 44.1 | 41.3 | 48.6 | 54.6 | 48.5 |
| Soil texture (USDA) | | Lo | SaClLo | ClLo | ClLo | ClLo | ClLo | Lo | SaLo | SaClLo |
| N Kjeldahl | (%) | 0.08 | 0.09 | 0.02 | 0.23 | 0.22 | 0.16 | 0.09 | 0.1 | 0.03 |
| P Olsen | (mg/Kg) | 66.1 | 54 | 9.07 | <5 | <5 | <5 | 80.9 | 56.4 | 10 |
| Potassium | (mg/Kg) | 288 | 263 | 129 | 205 | 219 | 164 | 291 | 242 | 137 |
| Magnesium | (mg/Kg) | 229 | 242 | 182 | 154 | 159 | 145 | 250 | 245 | 197 |
| Sodium | (mg/Kg) | 203 | 172 | 94 | 49 | 37 | 37 | 200 | 183 | 98 |
| C:N | ratio | 6.15 | 6.39 | 14.5 | 8.42 | 10.07 | 10.15 | 9.23 | 6.51 | 10 |
| Zinc | (mg/Kg) | 65 | 58 | 53 | 52 | 55 | 46 | 72 | 57 | 47 |
| Cadmium | (mg/Kg) | <0.5 | <0.5 | <0.5 | <0.5 | <0.5 | <0.5 | <0.5 | <0.5 | <0.5 |
| Chrome | (mg/Kg) | 26 | 28 | 27 | 44 | 48 | 37 | 31 | 27 | 23 |
| Lead | (mg/Kg) | 12 | 10 | 9.8 | 14 | 16 | 12 | 12 | 9.1 | 9.2 |
| Mercury | (mg/Kg) | <0.4 | <0.4 | <0.4 | <0.4 | <0.4 | <0.4 | <0.4 | <0.4 | <0.4 |
| Nickel | (mg/Kg) | 20 | 21 | 19 | 28 | 31 | 25 | 21 | 18 | 17 |
| Copper | (mg/Kg) | 45 | 30 | 38 | 21 | 23 | <20 | 63 | 34 | 27 |

**Table 5.** The pH and electrical conductivity values in the different amendment treatments. Letters indicate significant differences among amendment treatments and * indicates significant differences within sampling times. T0: moment of application of the amendments; T1: six months after the application of amendments. The letters A, B and C refer to the soil types. The treatments are indicated as CNT = control, DIG = digestate and COM = compost. Values are means with ± standard error. For each parameter in this table, *n* = 3.

| | pH | | EC (µS/cm) | |
|---|---|---|---|---|
| | T0 | T1 | T0 | T1 |
| A—COM | 8.30 ± 0.03 (b) | 8.32 ± 0.01 (ab) | 557.93 ± 37.71 (b) | 262.80 ± 12.99 (a) * |
| A—DIG | 8.27 ± 0.05 (b) | 8.24 ± 0.06 (b) | 592.03 ± 8.06 (b) | 322.47 ± 33.5 (b) * |
| A—CNT | 8.53 ± 0.05 (a) | 8.38 ± 0.03 (a) * | 303.50 ± 74.88 (a) | 257.03 ± 13.25 (a) |
| B—COM | 8.29 ± 0.15 (a) | 8.48 ± 0.08 (a) | 230.07 ± 7.30 (a) | 202.30 ± 23.75 (a) * |
| B—DIG | 8.07 ± 0.05 (a) | 8.41 ± 0.05 (a) * | 230.00 ± 19.87 (a) | 172.27 ± 25.55 (a) * |
| B—CNT | 8.10 ± 0.12 (a) | 8.53 ± 0.13 (a) * | 223.10 ± 11.99 (a) | 163.17 ± 7.24 (a) * |
| C—COM | 8.26 ± 0.04 (b) | 8.52 ± 0.01 (a) * | 476.27 ± 30.69 (b) | 334.90 ± 29.04 (a) |
| C—DIG | 8.10 ± 0.04 (c) | 8.25 ± 0.09 (a) * | 687.93 ± 20.18 (c) | 327.90 ± 1.54 (a) * |
| C—CNT | 8.49 ± 0.03 (a) | 8.43 ± 0.17 (a) | 324.30 ± 41.50 (a) | 179.90 ± 330.78 (a) * |

**Table 6.** *p*-Values of the rmANOVAs testing the effects of organic amendments, sampling times and their interaction in pH, EC and SOC. The letters A, B and C refer to the soil types. For each parameter in this table, *n* = 3.

|  |  | Amendment | Time | Amendment × Time |
|---|---|---|---|---|
| A | pH | 0.0004 | 0.0964 | 0.043 |
|  | EC | 0.0002 | 0.0001 | 0.0089 |
|  | SOC | 0.0008 | 0.042 | 0.3511 |
| B | pH | 0.2612 | <0.0001 | 0.0312 |
|  | EC | 0.2117 | 0.0005 | 0.1524 |
|  | SOC | 0.0003 | 0.0006 | 0.3695 |
| C | pH | 0.0045 | 0.0143 | 0.0244 |
|  | EC | 0.0407 | 0.0151 | 0.9501 |
|  | SOC | 0.0001 | 0.0008 | 0.0031 |

The effect of the amendments on soil organic carbon (SOC) also differs among the soils (Figure 3 and Table 6). In soil A, both amendments present a significant increase in the SOC after their application; this effect persisted after six months. In soil B, both amendments also show increased SOC after application; this effect can also be observed after six months, although the SOC concentration remains higher in the digestate treatment than the compost treatment. The SOC in soil C presents no significant effects of the amendments after application. However, after six months, SOC concentrations remain higher in the treatments with additional organic amendments.

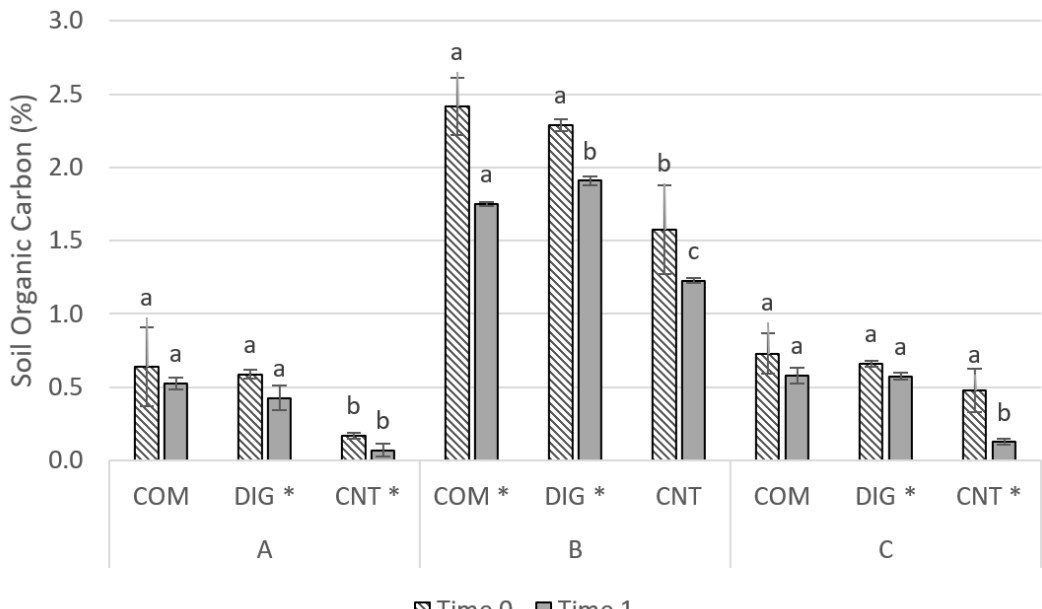

**Figure 3.** Soil organic carbon (SOC) in the different amendment treatments. Letters indicate significant differences among amendment treatments and * indicates significant differences in sampling times. Time 0: moment of application of the amendments. Time 1: six months after the application of amendments. Error bars indicate the standard deviation. The letters A, B and C refer to the soil types. The treatments are indicated as COM = compost, DIG = digestate and CNT = control. For each parameter in this graph, *n* = 3.

The statistical analyses show no significant differences between the organic amendment treatments in the soil resistance to penetration (Figure 4).

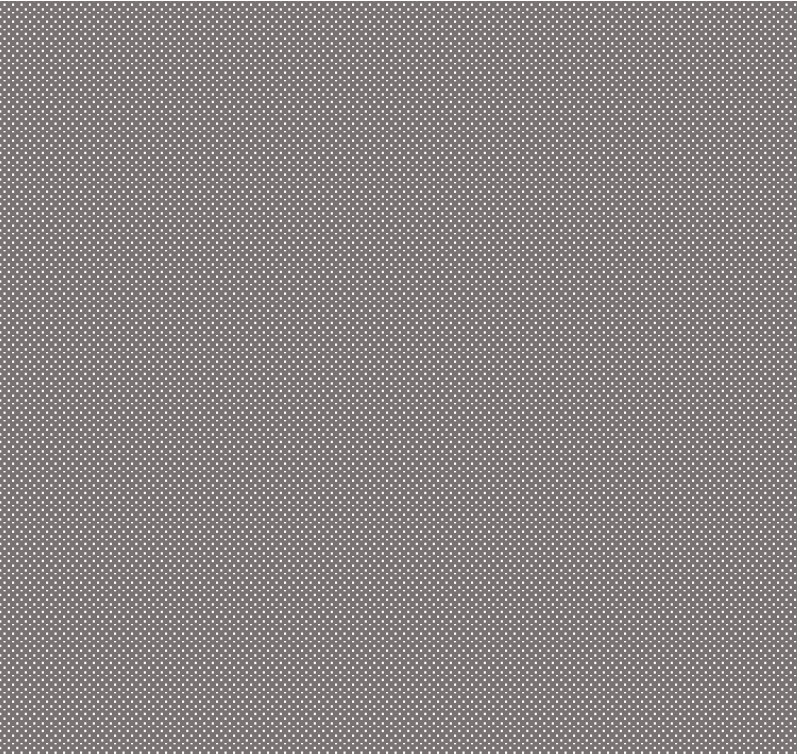

**Figure 4.** Box plots representing the resistance to penetration measured in the field. The letters A, B and C refer to the soil types. The treatments are indicated as CNT = control, COM = compost and DIG = digestate. For each parameter in this graph, *n* = 20.

In soil A, the organic amendment effect on aggregate stability differs among soil types and aggregate sizes (Figure 5). Both organic amendments significantly increase the stable aggregates of the highest diameter (>2 mm), and either compost or digestate also increase the stable aggregates of medium diameter as well (2–0.5 mm). In contrast, the control treatment has stabler aggregates of smaller diameters (0.5–0.053 mm). In soil B, however, the organic amendment treatments caused no significant effects on the aggregate stability.

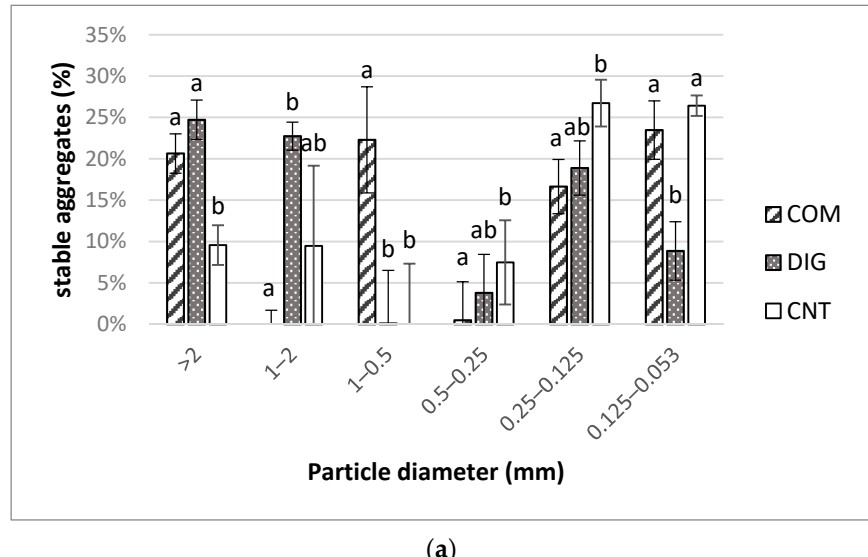

(a)

**Figure 5.** *Cont*.

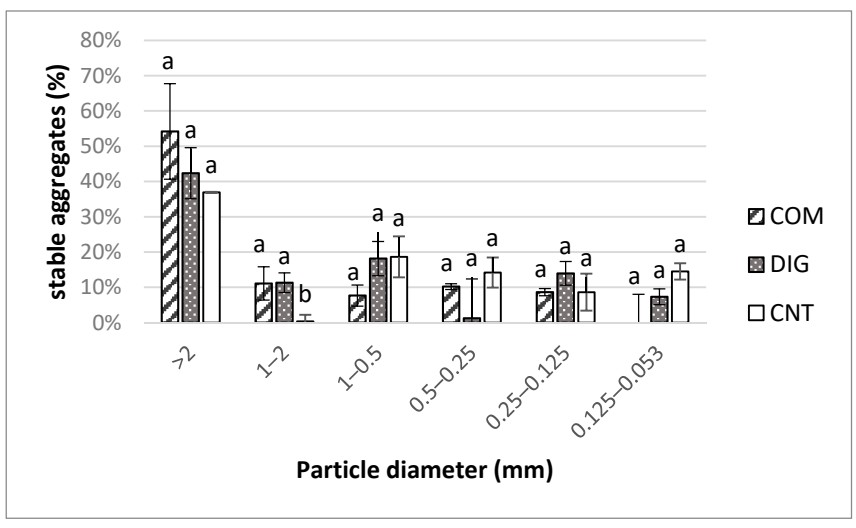

(**b**)

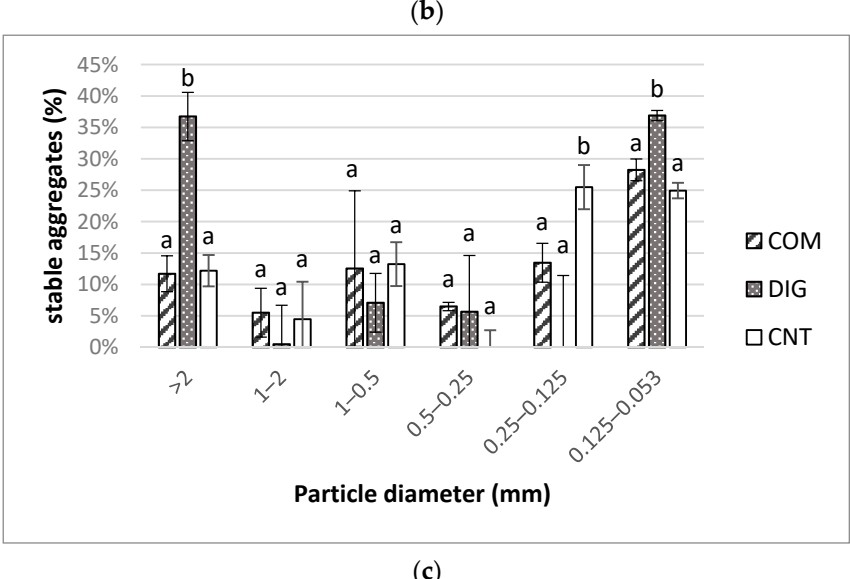

(**c**)

**Figure 5.** Proportion of stable aggregates of different sizes in the organic amendment treatments. (**a**) Proportion of stable aggregates in soil A; (**b**) Proportion of stable aggregates in soil B; (**c**) Proportion of stable aggregates in soil C. Treatments are indicated as CNT = control, COM = compost and DIG = digestate. The bars indicate the standard deviation. Different letters indicate significant differences among treatments for each soil type and particle size ($\alpha = 0.05$). For each parameter in this graph, *n* = 3.

Soil C shows a significantly increased stability of aggregates with the largest and smallest diameters (>2 mm and 0.125–0.053 mm) in the digestate treatment. However, the compost treatment did not show significant differences when compared to the control treatment.

## 4. Discussion

### 4.1. Soil Chemical Properties

One of the most evident effects of applying organic amendments was the increased level of macronutrient concentration in the constructed technosols. In the A and B technosols, the nitrogen concentration increased with the application of organic amendments (Table 4). At the same time, it must be noted that a proportion of the nitrogen supplied by the amendments could have been lost due to volatilisation under conditions of adequate moisture in the soil [22], since the pH was >8. Organic amendments tended to lower the soil pH, which is a relevant factor to consider when preventing potential N losses. A 10 cm



layer of topsoil was added to soil B, such that the organic amendments were added to the underlying soil layer. Therefore, the organic amendment did not have an immediate effect on the nutrient content of the soil surface layer, and their observed effect was limited in this soil.

According to Mikkelsen and Hartz [23], the C:N ratio of organic materials added as amendments is a good, but not an absolute predictor of whether N is immobilised (C:N > 25) or mineralised (C:N < 20). In all technosols studied, the soil C:N ratio had values below 20 (Table 3), and thus the nitrogen supplied by the amendments was most likely being mineralised and taken up by plants to a large extent [24].

The phosphorus content in the soil also increased with the applications of organic amendments (Table 4). Even though both organic amendments had the same origin, the phosphorus increases were greater in the compost treatment than the digestate treatment due to the mass loss during the composting process and the subsequent increase in the phosphorous concentration in the compost (Table 3). Phosphorus availability in plants is generally reduced in the pH range of the soils studied (between 8.07 and 8.53), thereby making the phosphorus applied with the amendments particularly important for early vegetation growth. It must also be considered that organic matter supplied by the amendments can reduce phosphorus fixation in soils [24]. Moreover, the amendment enhancement of the phosphatases activity and the physical decomposition of the supplied organic material may also result in greater phosphate mineralisation [25]. However, in the mid-term, low phosphorous availability is expected due to the alkaline pH of the highly calcareous materials used for technosol construction, which immobilises phosphorous as calcium triphosphate [26,27].

Potassium increased with the application of amendments (Table 4), also showing higher values in the compost treatment, which may also be linked to mass loss during the composting process. The amount of available potassium generally increases in organically amended soils due to the relatively high amount of potassium in organic amendments and the increased CEC related to the higher organic matter content [22]. Magnesium also increased with the application of organic amendments (Table 4). However, in this case, there were no remarkable differences between the two organic treatments. The calcium content of CEC was not considered relevant for this study, since the soils of the area were saturated with calcium due to the hyperabundance of calcite.

An increase in micronutrients and metals was also observed in the compost and digestate treatments (Table 4), corresponding to the amount of these elements in the organic amendments. The organic matter resulting from the amendments and its transformation into humus can increase the availability of micronutrients, such as zinc and copper, in the soil [22]. In calcareous soils, the pH, clay content, organic matter and CEC are the most significant factors of zinc adsorption [28]. As zinc availability decreases with increasing pH [28], the addition of organic amendments along with a decrease in soil pH (Table 5) increases the amount of available zinc [29].

Similarly, an important factor in the mobility of lead, copper, cadmium and zinc is the amount of organic matter in the soil [30]. Many authors [31–33] found that soils with higher organic carbon content associated with added organic fertilisers have a lower amount of available cadmium. This effect is explained by the high-cation-exchange capacity of organic matter and its ability to form stable complexes with cadmium. Nonetheless, cadmium complexation is not a relevant mechanism in this case, given the low concentration of cadmium in the resulting technosols after adding the organic amendments (Table 4) [34]. On the other hand, the mobility of metals is substantially reduced at the basic pH ranges of these soils.

Sodium also increased in the treatments with amendments (Table 4). Nonetheless, the technosols are saturated by calcium from the dissolution of carbonates (free calcium < 7000 mg/Kg in all soils), contributing to decreased sodium adsorption ratio (SAR) values [35]; therefore, sodicity is not a major issue on these soils. Regarding salinity, electrical conductivity increases with the application of the amendments (Table 5), and

soluble salts generally decrease significantly after six months, although some values remain high in the digestate treatments. In any case, the electrical conductivity values of such soils are far below the minimum, such that they cannot even be regarded as being slightly saline (2000 μS/cm) [35].

As previously mentioned, an important factor in the ability of plants to benefit from nutrients is the soil pH (Table 5). It should also be considered that the pH of the natural leptosols in the area, with values of 7.9 on average, is lower than those of the technosols. Nonetheless, higher pHs should not impact vegetation growth unless phosphorus is limited. In this regard, the organic amendment produced an initial acidifying effect in soils A and C, but not in soil B—an effect that may have been produced by the superficial layer of topsoil. This acidifying effect disappeared or attenuated over time, which can probably be attributed to the buffering capacity of these alkaline soils. The measured pH values (8.2–8.5) can reduce the availability of some nutrients, such as nitrogen and, particularly, phosphorus [22]. Therefore, the phosphorous contribution by the organic amendments become even more relevant at the pH ranges of calcareous soils.

As expected, after directly applying the organic amendments, the soil organic carbon increased and maintained a significantly high level after six months (Figure 3). In the soils without topsoil, the amendment-induced improvement of soil aggregation may have contributed to the physical protection of soil organic carbon against microbial decomposition and soil carbon stabilisation [36,37]. Soil structure can contribute to organic matter storage by providing physical protection for organic substrates, thereby preventing their degradation by microbial communities. This mechanism is mediated by both pore-size distribution and the formation of soil aggregates [38].

### 4.2. Soil Physical Factors

The surface horizons of the technosols tended to resemble the A horizons of the natural haplic leptosols of the Garraf area in terms of textural classes, with textures from loam to clay loam, although some of the plots contained a slightly higher proportion of sand and reached sandy clay loam textures (Table 4). One of the main differences with natural soils is the higher proportion of stones in these technosols, i.e., 15–35% in natural soils compared to 60–75% in technosols. These coarse elements also presented very wide size ranges, including gravel diameter of a few centimetres and blocks that can reach up to 50 cm in diameter. This generates a great internal heterogeneity in the technosols, which causes high dispersion in some variables, as can be seen in the bulk density measures (Table 4).

The resistance to the soil surface penetration (Figure 4) informs whether clayey textures and soil compaction could be a problem for initial plant development. Soil A treated with digestate was the only one that showed signs of compaction. However, this was most likely due to the accidental compaction by the machinery used to build the plots and was not related to the organic amendment treatments.

In this study, the addition of digestate increased the amount of macroaggregates in soils A and C (Figure 5), which may be related to a higher biological activity compared to controls and compost treatments. The stability of soil aggregates depends on both abiotic and biotic factors, the former being mainly related to the clay content of the soil [39]. The aggregation process occurs in a hierarchical way, whereby small soil particles stabilise into mature soil aggregates when biotic factors are the main aggregating agents [39]. Although this is specific to each soil, the hierarchy of soil aggregates can be classified in three main orders: clay microstructures (<2 μm in diameter), microaggregates (2 to 250 μm) and macroaggregates (>250 μm) [39]. In clayey microstructures, clay–humic complexes are stabilised by humic acids and bivalent inorganic ions (e.g., $Ca^{2+}$). Microaggregates are directly stabilised by microbial products, such as polysaccharides, hyphae and bacterial colonies [39]. In contrast, the formation of macroaggregates and their transient stabilisation is mediated by the presence of plant roots, fungal and soil fauna activities [40], and are thus more vulnerable to disturbances. However, the effect of added organic amendments was

not observed in soil B, where the topsoil probably retained part of the edaphic structure and biota from the former natural soils of the area, and the amendments did not have as much impact as on the other soils. The treatments in soil B only showed a significant improvement in the 1–2 mm fraction of aggregates.

## 5. Conclusions

The application of organic amendments was particularly suitable for improving the properties of technosols constructed without scrapped topsoil, such as nutrient concentration, soil organic matter and aggregate stability, thereby providing them strong robust structures. As expected, soil organic carbon increased after direct application of both types of organic amendments, and the effects persisted after six months. Macroaggregate stability had a greater improvement with the application of digestates compared to compost; this is probably due to the enhancement of biological activity. On the other hand, organic amendments also increased the EC and lowered the soil pH right after their application; six months after applying the amendments, these effects only persisted in the case of digestates.

It must also be noted that the application of organic amendments did not significantly improve most of the superficial soil properties when topsoil was applied as a superficial layer of technosols. In this case, the technosols constructed using topsoil already had high soil organic carbon content, good aggregation and structure as well as adequate nutrient content, even without amendments. Therefore, scrapped topsoil should be appropriately stored during quarry exploitation and used in restoration works whenever possible. However, as topsoil is not always available or, at least, not in sufficient quantities, the use of mineral and organic amendments, and particularly digestate from source-separated organic household waste, may be the most beneficial treatment for improving soil properties and soil carbon sequestration for the ecosystem restoration of Mediterranean habitats. Moreover, this option can be very helpful in contributing towards circular economies and waste valorisation, since urban mineral and organic wastes are used as resources for technosol construction.

**Author Contributions:** Conceptualisation, V.C. and J.M.A.; methodology, V.C. and J.M.A.; validation, V.C. and J.M.A.; formal analysis, P.S.; investigation, P.S., V.C., J.M.A. and D.F.; resources, J.M.A., V.C. and I.R.; data curation, S.M.-J., V.C. and P.S.; writing—original draft preparation, P.S.; writing—review and editing, P.S., S.M.-J., V.C., D.F., I.R., J.M.A., M.P. and R.G.; visualisation, P.S.; supervision, M.P., V.C. and J.M.A.; project administration, J.M.A.; funding acquisition, M.P., J.M.A. and V.C. All authors have read and agreed to the published version of the manuscript.

**Funding:** This paper has been produced thanks to the funding of the project "Research, innovation, and promotion of the use of organic amendments and organic waste for soil restoration" (ref. G0832_2021_38), financed by the Catalan Waste Agency (Government of Catalonia).

**Data Availability Statement:** The data presented in this study are available on request from the corresponding author. The data are not publicly available due to privacy reasons.

**Acknowledgments:** The authors acknowledge the PROMSA company and the CONSORCI PER A LA GESTIÓ DELS RESIDUS DEL VALLÈS ORIENTAL for their support on the construction of the experimental plots.

**Conflicts of Interest:** The authors declare no conflict of interest. The funders had no role in the design of the study; in the collection, analyses, or interpretation of data; in the writing of the paper; or in the decision to publish the results.

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
