# Peer review of "Physical and Chemical Properties of Limestone Quarry Technosols Used in the Restoration of Mediterranean Habitats"

_land, doi:10.3390/land12091730_

Round 1

Reviewer 1 Report

I find the paper to be intriguing, addressing a crucial subject. The significance of engineered/manufactured soils for restoring areas with slow soil formation cannot be understated.

Overall, the paper possesses a straightforward and coherent narrative, making it easy to comprehend and follow.

I will begin by highlighting a few major concerns, followed by some minor observations.

Firstly, the study's specific implications and general impact need clearer articulation. The Discussion and Conclusion sections should delve deeper into the contribution this paper makes to the field. What distinguishes this work within the discipline?

The Methods section requires further elucidation in certain areas, primarily to ensure readers can both interpret the data based on the methods employed and replicate the study. These clarifications will aid in the comprehension and reproducibility of the research.

While the majority of the language is used effectively, there is potential for improvement through the review of an English Language editor.

Here are specific line-by-line comments:

  • Line 33: Can you provide an explanation of the term "environmental regulations"?
  • Line 39: The relief will always be considered unnatural due to its (re)constructed nature.
  • Line 42: Consider changing "recover" to "restore."
  • Line 45: How is the proper preservation of these elements achieved?
  • Line 52: It is important to distinguish "excavated soils" from manufactured/engineered soils, which are often blended off-site and transported to urban green infrastructure projects.
  • Line 60: "Coarse elements" might be better phrased as "coarse fraction."
  • Line 71: You mention "one of the main factors," but several factors are listed prior. Clarify which factor you refer to here.
  • Line 77: Could you provide more specificity regarding the "deficiency in physical properties"?
  • Line 84: Consider addressing the role of soil biology in utilizing organic amendments in technosols.
  • Line 90: Could you elaborate on the "several experiments" or direct readers to a suitable reference?
  • Line 91: Replace "stablished" with "established."
  • Line 92: While the map is informative, an accompanying satellite image of the site could provide contextual understanding.
  • Line 95: Does "scarcity of soil" refer to complete absence or very shallow soil?
  • Line 110: How was the topsoil stored?
  • Line 113: It would be valuable to describe the ecosystems prior to the quarry site. This could serve as a baseline for understanding site history and landscape restoration potential.
  • Line 126: Can you provide justification for the specific sampling depths chosen?
  • Line 137: How long after the establishment of treatments were the plants sown?
  • Line 143: Explain the rationale for exclusively sampling the upper 20 cm; a more comprehensive profile sampling might be warranted.
  • Line 152: Clarify the meaning of "together with other measurements."
  • Line 156: The method for bulk density measurement is unique; including SOPs or references explaining the method would be beneficial.
  • Line 173: Are the results presented in the first paragraph of the Results section statistically significant?
  • Table 1: Include the n number in this and all data figures.
  • Table 6: Specify units for the measured variables.
  • Line 202: The information in this paragraph contradicts the results shown in Figure 3. For instance, you state a "significant increase of SOC after application" in Soil A, while Figure 3 depicts a decrease between Time 0 and Time 1.
  • Figure 3: There appears to be an error with the Y-axis. The repeated values should be addressed.
  • Figure 3: Since the % sign is displayed in the Axis title, it need not be repeated after each point on the Y-axis.
  • Figure 3: Consider using labels like "On Application" and "6 months post-application" instead of "Time 0" and "Time 1."
  • Figure 5: Rectify the order of X-axis categories, ensuring consistency in presentation.
  • Line 257: How would you propose testing this aspect within your study?
  • Line 286: Specify the authors when mentioning "many authors"; provide citations here.
  • Line 319: Be cautious about suggesting carbon storage saturation without conducting further experiments to test this hypothesis.
  • Line 332: Acknowledge the feedback loop between soil structure and SOC; they mutually influence each other.
  • Line 324: Consider using "community" instead of "enzymes."
  • Line 335: Elaborate on the phrase "from which it has not been possible to draw significant conclusions."
  • Line 339: How were signs of compaction determined to be "clear"?

While the majority of the language is used effectively, there is potential for improvement through the review of an English Language editor.

Reviewer 2 Report

The manuscript is worth publishing, I have a few minor comments that need to be addressed (if possible):

1. Lines 261-267, need more clarity, somehow the message is not clear.

2. Lines 327-335, the authors need to find a way to show the dispersion they have discussed, I believe Table 4 is not enough for the purpose.

3. Line 266-267, elaborate please, so that your assertion may be more clear to readers. 

What about Calcium phosphate precipitates in calcareous soils?

A thorough recheck for typo, grammatical and syntax is recommended.

Dear Editor, 

The paper is worth publishing. Overall, the subject matter and ideas discussed in the paper are interesting. The MS is well formulated and may be essential in terms of scientific inquiry. I have made some comments that do not bear any impact on the quality of the MS, and if the authors deem it possible may be incorporated. 

Regards

Round 2

Reviewer 1 Report

Thank you for digesting and addressing the issues I raised in my peer review. I am happy with you responses and revisions, and I believe this paper now to be of publishable quality. 

I suggest that the authors work with the journal’s language editors to ensure accuracy

Author Response

Thank you for your comments.

We have had the english revised and provide you with a new version.